# An Overview of Clinical Manifestations of Dermatological Disorders in Intensive Care Units: What Should Intensivists Be Aware of?

**DOI:** 10.3390/diagnostics13071290

**Published:** 2023-03-29

**Authors:** Ali Al Bshabshe, Wesam F. Mousa, Nashwa Nor El-Dein

**Affiliations:** 1Department of Medicine/Adult Critical Care, King Khalid University, Abha 61413, Saudi Arabia; 2College of Medicine, Tanta University, Tanta 31512, Egypt

**Keywords:** dermatological disorders, cutaneous manifestations, skin diseases, local signs and symptoms, intensive care unit, ICU, critical care unit, intensivists, systemic disease, drug reaction

## Abstract

Acute skin failure is rarely the primary diagnosis that necessitates admission to an intensive care unit. Dermatological manifestations in critically ill patients, on the other hand, are relatively common and can be used to make a key diagnosis of an adverse drug reaction or an underlying systemic illness, or they may be caused by factors related to a prolonged stay or invasive procedures. In intensive care units, their classification is based on the aetiopathogenesis of the cutaneous lesion and, in the meantime, distinguishes critical patients. When evaluating dermatological manifestations, several factors must be considered: onset, morphology, distribution, and associated symptoms and signs. This review depicts dermatological signs in critical patients in order to lay out better recognition.

## 1. Introduction

Dermatological manifestations (DMs) in critically ill patients are receiving increased attention in intensive care medicine in order to obtain an accurate diagnosis at the right time. Without appropriate knowledge and greater awareness of the various pathophysiological aspects and clinical presentations of skin diseases, their interpretation is typically difficult for intensive care physicians [1,2,3]. 

DMs should be taken very seriously when they occur suddenly in critically ill patients, especially when they are receiving newly introduced medications [4]. Identifying DMs can also aid in the detection of an underlying disease [5,6]. Skin manifestations in meningococcal sepsis [7] and ecthyma gangrenosum in Pseudomonas aeruginosa infection [8,9] are two examples. Other times, DMs reflect secondary effects of devices or procedures or they are simply due to the patient’s critical condition [1]. Nonetheless, DMs can present in a variety of ways, and a definitive diagnosis frequently necessitates the collaboration of a dermatologist [10]. 

Skin examination should be made a routine part of ICU care [11]. An algorithm [12] proposed to identify the etiology of DMs in ICUs suggested first to identify DMs, then, as a second step, to differentiate DMs that appear before or after ICU admission and, finally, to determine the need for consulting a dermatology specialist. The purpose of this review is to categorize and describe the most common DMs found in critically ill patients.

## 2. Classification of DMs in Intensive Medicine 

We tweaked a classification developed in an earlier study [13]. DMs in ICUs were classified into four groups: life-threatening DMs; DMs associated with adverse drug reactions; DMs associated with systemic diseases; DMs as a consequence of critical illness. (all summarized in Appendix A).

## 3. Group 1: DMs Denoting Life-Threatening Skin Diseases That Require ICU Admission

They are rare but they are frequently severe and have a high mortality rate. Rapid deterioration of cutaneous functions will result in fluid, protein, and electrolyte loss, as well as thermoregulatory impairment, hypercatabolic state, and increased infection risk [3]. 

Well-known, potentially life-threatening DMs disorders are:

### 3.1. Pemphigus Vulgaris (PV) 

This is more common in middle-aged or older people. It is distinguished by generalized pruritic urticarial plaques, fragile subepithelial blisters, and skin and mucus membrane sores [14]. 

It is an autoimmune disease in which the body’s immune system incorrectly recognizes skin proteins as foreign and thus produces antibodies to attack the foreign protein [15,16]. Trigger factors include medications [17] such as antibiotics, calcium channel blockers, ACE inhibitors, NSAIDs, salicylates, and interleukins, vaccines such as influenza [18,19], swine flu [20], tetanus toxoid [21], and COVID-19 vaccines [22,23,24], viral infections such as herpes simplex [25], hepatitis B and C [26,27,28], bacterial infections such as Helicobacter pylori [29], parasitic infections such as Toxoplasma gondii [30,31], and following organ transplantation [32,33]. Physical factors such as trauma, surgical interventions, thermal or electric burns, ultraviolet exposure, radiotherapy, and photodynamic therapy can also cause PV [34]. 

A biopsy from the lesion confirms the diagnosis, revealing acantholysis and IgG deposition between acanthocytes by direct immunofluorescence. Serum tests detect IgG via indirect immunofluorescence or autoantibodies via ELISA [35,36]. The current cornerstone treatment for pemphigus is local and systemic glucocorticoids [37]. Other immunosuppressants such as azathioprine and methotrexate are also standard options for the treatment of pemphigus patients [28]. Furthermore, the novel use of a CD20 monoclonal antibody has been successful, and newly developed biological agents may provide more effective therapies for pemphigus patients [38,39]. 

### 3.2. Stevens–Johnson Syndrome (SJS) and Toxic Epidermal Necrolysis (TEN) 

It occurs in patients of all ages but particularly in children, young adults, and the elderly above the age of 80 [40,41]. It is distinguished by rapidly progressing painful symmetric ill-defined erythematous lesions with central dusky purpura or bullae that coalesce. Within days, widespread epidermis shedding occurs, resulting in fluid loss and an increased risk of infection. Toxic epidermal necrolysis is a more severe reaction that causes full-thickness epidermal necrosis and exfoliation [41]. 

Many cases are drug-induced hypersensitivity reactions, with common triggers including NSAIDs, allopurinol, anticonvulsants (such as lamotrigine, phenytoin, and carbamazepine), sulfonamides, and nevirapine [42]. Mycoplasma pneumoniae infection is the second most common trigger, but in more than one-third of cases, no trigger is found [43]. 

A physician can usually diagnose TEN after taking a history and performing a physical examination, but in some cases, a skin biopsy may be required to confirm the diagnosis. Treatment focuses on removing the trigger, caring for lesions, and managing pain. For severe eye lesions, systemic glucocorticoids are recommended [44]. More multicentric studies are needed to provide a reliable answer about the efficacy of other treatment modalities such as intravenous immunoglobulins (IVIGs), cyclosporin, N-acetylcysteine, thalidomide, infliximab, etanercept, and plasmapheresis in the treatment of SJS/TEN [45]. 

### 3.3. Necrotizing Fasciitis 

It occurs in patients of all ages but is more common in the elderly. It is distinguished by a severe, insidiously evolving soft-tissue infection of the fascia that moves along the fascial plane, with secondary necrosis of the subcutaneous tissues [46]. 

Although it is a polymicrobial infection, group A beta-hemolytic streptococci are the organisms most closely associated with necrotizing fasciitis. It can also be idiopathic or develop as a complication of a variety of surgical procedures [47]. 

Examination, bloodwork for signs of infection and muscle damage, CT scan, MRI, and ultrasound of the affected area are used to diagnose necrotizing fasciitis. However, surgical debridement with microbiological and histopathological examination of soft tissue is the gold standard action for both diagnosis and treatment. Aerobic coverage with ampicillin–sulbactam or carbapenems should be combined with anaerobic coverage with metronidazole or clindamycin [48,49]. 

### 3.4. Toxic Shock Syndrome (TSS) 

It can affect people at any age or any gender. It is a soft tissue infection that occurs due to toxins produced after Staphylococcal or, less commonly, streptococcal bacterial infections usually in the setting of menstruating women using tampons, post-surgical infections, skin wounds, burns, retained nasal packing, and dialysis catheters [50,51] 

It is distinguished by a generalized sunburn-like rash, particularly on the palms and soles, as well as mucous membrane hyperemia (vaginal, oropharyngeal, or conjunctival). It causes headaches, fever, myalgia, hypotension, and confusion as well as creatine phosphokinase levels that are more than twice the normal level. A devastating immune response to the infection may result in septic shock [52,53]. 

Swabs from the vagina, cervix, and throat are collected, along with blood and urine samples, for culture and sensitivity testing. Until culture and sensitivity results are available, the antibiotics of choice are oxacillin and first-generation cephalosporin [52,54]. 

### 3.5. Staphylococcal Scalded Skin Syndrome (SSSS) or Ritter’s Disease 

It is a staphylococcus aureus skin infection in which the skin is scalded as if burned by hot liquids, but unlike TSS, it does not have systemic manifestations. The majority of cases occur in children under the age of two, with a mortality rate of around 4%, while critically ill adults have a mortality rate of more than 60% [55]. 

A skin biopsy can confirm the diagnosis. Cefazolin, nafcillin, oxacillin, vancomycin, daptomycin, and linezolid are common antibiotics used to treat staph infections [55,56]. 

### 3.6. Meningococcemia 

Meningitis occurs when meningococci infect the meninges, and meningococcemia occurs when they remain in the blood. The prevalence is higher at the extremes of age, and mortality can reach 80% without intervention and 30–50% with it [57]. 

It is distinguished by a petechial rash on the legs and trunk that rapidly disseminates, turning purpuric necrotic and growing into large patches similar to skin bruises. Patients may present with meningitis, meningitis with meningococcemia, or only meningococcemia [7]. 

The definitive diagnosis is the presence of meningococci in the blood, CSF, throat swab, or skin biopsy from the rash. Furthermore, CSF analysis typically reveals high protein, low glucose, CSF with a glucose to blood glucose ratio of 0.4 or lower, high lactate, and high neutrophils [58,59,60,61]. Antimicrobials most commonly used to eradicate meningococci include penicillin, rifampin, ciprofloxacin, and ceftriaxone [62,63,64]. 

### 3.7. Rocky Mountain Spotted Fever 

It is a rickettsial disease affecting any age with an overall mortality rate up to 25% in untreated cases and 3–5% in treated cases. In a patient with a history of tick bite or exposure, the rash begins as erythematous macules on the wrists, hands, and feet and quickly spreads to the trunk and face [65,66]. 

Early detection and treatment are essential for lowering morbidity and mortality. The serological detection of anti-rickettsial antibodies is the standard and most commonly used test for confirmation of the diagnosis. PCR testing is quick and allows for species identification. Doxycycline is the preferred treatment for both adults and children [67,68,69]. 

### 3.8. Exfoliative Erythroderma

It is distinguished by diffuse erythema and scaling of 90% of the body surface area. It is caused by a variety of underlying cutaneous disorders, drugs, and malignancies, though some cases are idiopathic. It can be fatal [70,71,72]. 

Erythroderma is diagnosed through the patient’s history and examination. Treatment includes fluid replacement for insensible losses, correction of electrolyte and thermoregulatory disturbances, and the administration of antihistamines and glucocorticoids [73]. 

### 3.9. Generalized Pustular Psoriasis (GPP)

It is an immune-mediated systemic skin disorder exacerbated by viral or bacterial infections, corticosteroids, psychological stress, hypocalcemia, and pregnancy [74]. The incidence is highest between the ages of 40 and 59. It is distinguished by the widespread eruption of sterile pustules, which may be accompanied by generalized symptoms of pain, fever, malaise, or fatigue [75,76]. 

The history and examination of the patient are usually sufficient for diagnosis and treatment initiation; however, skin biopsy may be required to determine the type of psoriasis and rule out other disorders [77,78]. Topical and phototherapy retinoids, methotrexate, cyclosporine, and corticosteroids are used in treatment. TNF-alpha inhibitors can be used, but they should be combined with methotrexate to avoid the formation of anti-drug antibodies [75]. Guselkumab, risankizumab, and tildrakizumab are IL-23 inhibitors that represent a new class of biological therapy for the treatment of psoriasis. They have been shown to be safe and effective, with no serious side effects reported even in frail patients such as the elderly and patients with comorbidities [79]. Data from patients with COVID-19 show that biologics may cause a mild psoriasis exacerbation in 10% of cases and that treatment discontinuation should be avoided [80]. 

## 4. Group 2: DMs Associated with Adverse Drug Reactions

Up to 10% of hospitalized patients may experience drug-induced skin reactions (DISRs). The majority of DISRs manifest as a self-resolving maculopapular rash, urticaria, angioedema, and erythema multiforme. Life-threatening patterns such as erythroderma, Stevens–Johnson syndrome, and toxic epidermal necrolysis are less common [81,82]. Almost any drug can cause DISRs, but antibiotics, NSAIDs, antiepileptics, and contrast media are frequently involved [83]. 

Earlier, six types of adverse drug reactions were identified: [84] type A: dose-related (augmented), type B: non-dose-related (bizarre), type C: dose-related and time-related (chronic), type D: time-related (delayed), type E: withdrawal (end of use), and type F: failure of therapy (failure). Adverse drug reactions are a significant cause of morbidity and mortality [85], as well as an increase in healthcare costs [86]. A significant number (62.8%) of adverse drug reactions are preventable [87], and successful management requires early detection and prompt intervention with antihistamines, epinephrine, and steroids [88]. 

## 5. Group 3: DMs Associated with Systemic Diseases 

Any skin signs that may indicate underlying systemic disease should be noted by a dedicated intensivist. This review does not aim to cover all systemic diseases with potential DMs, but rather to highlight the most common.

### 5.1. Seborrheic Dermatitis

It is a chronic superficial inflammatory disease that affects the skin’s sebaceous glands, causing itchy, thin, fine erythematous, and scaly plaques on the scalp, face, ears, and chest [89]. It can be caused by systemic diseases such as Parkinsonism [90], HIV infection [91], or intracranial hemorrhage [92]. 

### 5.2. Erythema Multiforme

It is a type of cutaneous hypersensitivity reaction that is usually caused by a viral, bacterial, fungal, or parasitic infection, but can also be caused by drug sensitivity. It can also indicate collagen diseases, vasculitides, non-Hodgkin lymphoma, leukemia, multiple myeloma, myeloid metaplasia, and polycythemia in rare cases [93]. It is distinguished by symmetrical macules, papules, plaques, vesicles, or bullae, usually with an iris appearance, that begin on the extremities, particularly on extensor surfaces, and may extend to mucosal surfaces. It could be mild (erythema multiforme minor) or severe (erythema multiforme major) [94]. 

### 5.3. Vitiligo 

Depigmented skin and white hair patches tend to grow larger over time. It can affect any part of the skin, but it is most noticeable on the face, hands, and wrists, as well as around body orifices. Although specific causes are unknown, studies indicate a link to immune system changes [95]. The condition is frequently associated with autoimmune thyroid disease [96], pernicious anemia [97], diabetes mellitus [98], systemic lupus erythematosus [99], celiac disease [100], and Addison’s disease or after using biological agents such as adalimumab, infliximab, ustekinumab, secukinumab, and ixekizumab [101]. 

### 5.4. Erythema Nodosum

Panniculitis causes painful, erythematous nodules on the shins and, on rare occasions, elsewhere. It typically affects women between the ages of 20 and 40 [102]. It is idiopathic in 50% of cases but can reflect a variety of viral, bacterial (especially streptococcal pharyngitis), and fungal infections, drug sensitivity (especially sulfonamides and salicylates), hepatitis B vaccination, oral contraceptives as well as inflammatory bowel disease (Crohn’s disease, sarcoidosis, and Behçet’s disease), non-Hodgkin’s lymphoma, carcinoid tumors, and pancreatic cancer [103]. 

### 5.5. Cutaneous Metastases

Any malignant neoplasm can rarely spread to the skin. They typically manifest as solitary or multiple discrete firm painless fleshy nodules that emerge rapidly in close proximity to the primary neoplasm. The most common primary tumors with cutaneous metastases are those of the breast, lung, large intestine, and kidney [104,105]. 

### 5.6. Paget’s Disease

Unilateral eczematous plaque of the nipple and areola, surrounded by a darker circle of skin. It is strongly associated with an underlying invasive breast carcinoma or intraductal carcinoma [106].

### 5.7. Acanthosis Nigricans

It is a dark discolored hyperkeratotic velvet-like skin condition with plaques in body folds and creases (e.g., groin, axillae, neck) [107]. It is usually associated with hyperinsulinemia (present in obesity and type 2 diabetes mellitus), the use of systemic corticosteroids, nicotinic acid, diethylstilbestrol, and isoniazid, or it can be an indicator of underlying adenocarcinoma, particularly gastrointestinal, where it has a sudden onset and a wide distribution that includes the face, palms, and trunk [108]. 

### 5.8. Sweet’s Syndrome (Acute Febrile Neutrophilic Dermatosis)

Painful cutaneous and mucosal erythematous to violaceous plaques on the face, arms, neck, and trunk, as well as fever, malaise, arthralgias, myalgias, and conjunctivitis, typically affecting middle-aged women [109]. It may be idiopathic, but it has been linked to acute myelocytic or myelomonocytic leukemia, as well as inflammatory bowel disease, rheumatoid arthritis, and pregnancy [110,111]. 

### 5.9. Amyloidosis of the Skin

Because of increased blood vessel fragility caused by amyloid infiltration of the vessels, papules on the eyelids and extremities that become purpuric and ecchymotic after pressure or rubbing develop. It could be a pemphigus symptom of multiple myeloma [112]. 

### 5.10. Erythema Gyratum Repens

Characteristic pruritic concentric erythematous bands forming a wood-grain pattern. It has a strong link to lung cancer and, to a lesser extent, breast, cervical, and gastrointestinal cancers [113,114]. 

### 5.11. Cyanosis 

The skin and mucus membranes are discolored in blue. In the case of central cyanosis, it is best appreciated on the tongue, oral mucosa, and conjunctiva, and in the case of peripheral cyanosis, it is best appreciated on the periphery. Central cyanosis indicates desaturated hemoglobin, whereas peripheral cyanosis may indicate desaturated hemoglobin or decreased tissue perfusion. Central cyanosis in an intensive care unit can be caused by pneumonia, massive pleural effusion, or pulmonary embolism. Isolated peripheral cyanosis could be caused by cold extremities, vascular disease, vascular obstruction, left ventricular failure, or shock [115]. 

### 5.12. Clubbing 

Bulbous, club-like deformation of the distal parts of the fingers and toes caused by connective tissue proliferation. Common causes include cyanotic heart disease, benign mesothelioma, infective endocarditis, cystic lung fibrosis, lung cancer, lung abscess, bronchiectasis, and ulcerative colitis [116,117,118]. 

### 5.13. Necrolytic Migratory Erythema (Glucagonoma Syndrome)

It occurs in conjunction with a pancreatic islet alpha-cell tumor (glucagonoma). It can also occur in hepatic disease and malabsorption syndrome (pseudo glucagonoma syndrome) [119,120]. The sacrum, perineum, buttocks, intertriginous and periorificial areas, distal extremities, and face are all affected by a radially distributed, extremely erythematous rash with scaly necrotic plaques. It is commonly associated with diabetes symptoms, diarrhea, and weight loss [121,122]. 

### 5.14. Cutaneous Leukocytoclastic Vasculitis

It occurs as a result of vasculitis, resulting in palpable purpura on the lower extremities that may be burning, painful, or pruritic. It could be caused by medications, infection, collagen-vascular disorders, liver cirrhosis, hematologic malignancies, COVID-19 vaccination, or idiopathy [123,124]. 

### 5.15. Lichen Planus 

It is a cutaneous and mucous membrane inflammatory disorder characterized by pruritic, violaceous flat papules on the wrists, medial thighs, genitalia, lower back, ankles, and oral mucosa. They are surrounded by lattice-like white lines that are most visible on the oral mucosa, where erosions can also be seen. It can be idiopathic, drug-induced, or caused by biliary cirrhosis, hepatitis B virus immunization, or hepatitis C infection. Patients who take vitamin D supplements as an adjunct to conventional steroid therapy experience significant improvement in their symptoms [125,126,127]. 

### 5.16. Necrolytic Acral Erythema

Pruritic keratotic plaques on the upper and lower extremities. It was previously identified as a distinct finding in hepatitis C infection, but it was also seen after SARS-Cov-2 infection and hypothyroidism without associated hepatitis C virus infection [128,129,130]. 

### 5.17. Hereditary Hemorrhagic Telangiectasia (Osler-Weber-Rendu Syndrome)

It is a hereditary condition characterized by multiple hemorrhagic mucocutaneous telangiectasias on the skin and mucus membranes that represent small arterio-venous malformations. It is accompanied by frequent epistaxis and GIT bleeds, resulting in iron deficiency anemia. Furthermore, those patients are at risk of a potentially fatal hemorrhage involving the lungs, liver, and brain [131,132]. 

### 5.18. Pyoderma Gangrenosum

Ulcers that are painful and have boggy, blue/purple undermined borders. All parts of the body can be affected, but the legs are the most commonly affected. Ulcers frequently develop as a result of trauma and begin as pustules or nodules that ulcerate and spread centrifugally. Patients with underlying rheumatoid arthritis, inflammatory bowel disease, blood dyscrasias, or idiopathic rheumatoid arthritis are more likely to be affected [133,134]. 

### 5.19. Nephrogenic Systemic Fibrosis

It occurs in patients on dialysis who have end-stage renal disease. Induration, thickening, and hardening of the skin on the extremities and trunk. The disease typically progresses, resulting in joint contractures [135]. 

### 5.20. Diabetes-Mellitus-Related Skin Conditions

Bacterial infections, fungal infections, diabetic dermopathy, necrobiosis lipoidica diabeticorum, diabetic blisters, and eruptive xanthomatosis are examples of these [136]. 

### 5.21. Pityriasis Rosea

Acute self-limiting pruritic papulosquamous oval patches typically appear in a Christmas-tree pattern on the trunk and proximal extremities, with mucosal involvement in 16% of cases. It primarily affects children and young adults, with a typical course of 6 to 8 weeks. It is frequently preceded by an initial oval spot known as the herald patch 2 weeks before. The condition has clinical variants that can make diagnosis difficult, especially in the absence of the herald patch. Atypical morphologies include vesicular, purpuric, urticarial, generalized popular, lichenoid, erythema multiforme-like, and follicular. It may also occur unilaterally, localized, acral, or as a limb girdle [137,138]. Although the aetiopathogenesis of the disease is unknown, a large body of evidence using PCR tests and other modern biological techniques has revealed a close relationship between PR and HHV6 and/or HHV7 systemic active infection [139].

Reassurance and symptomatic treatment are typically used. Emollients, antihistaminics, topical steroids, and macrolides (particularly erythromycin) are the standard treatments. Acyclovir can be used in relapsing cases or those that occur during early pregnancy [137,138]. Recently, however, it was reported that COVID-19 might cause atypical forms of pityriasis rosea refractory to conventional therapies [140]. 

## 6. Group 4: DMs as a Consequence of the Critical Illness

### 6.1. Skin Infections

Because of their prolonged immobility, malnutrition, decreased immunity, inadequate skin hygiene, hyperpyrexia, and device-insertion-related skin injuries, ICU patients are predisposed to skin infections [141]. Among the most common skin infections are: Cellulitis is an infection of the dermis and subcutaneous tissue with poorly defined borders caused by Streptococcus or Staphylococcus species [142].Erysipelas is a superficial form of cellulitis with well-defined borders that is almost always caused by Streptococcus [143].Impetigo is the lifting of the stratum corneum that results in bullae caused by Streptococcus or Staphylococcus [144].Folliculitis is an inflammation of the hair follicles that is most commonly caused by Staphylococcus aureus. If the follicle infection becomes deeper and involves more follicles, it progresses to the furuncle and carbuncle stages [145].Candidiasis is a yeast infection of the skin and mucous membranes caused by Candida albicans [146].

### 6.2. Dermatological Disorders Induced by Vasopressor Drugs

Vasopressor extravasation can result in serious complications ranging from simple local reactions to skin necrosis and compartment syndrome [147,148]. Vasopressors can also contribute to the development of pressure ulcers in critically ill patients [149]. 

### 6.3. Pressure Sores

Even with best practices such as pressure redistribution equipment, intensive physiotherapy, and protective dressings, critically ill patients are at risk of developing pressure sores as a result of their prolonged stay, immobility, deep sedation, vasoactive drug use, hypotension, anasarca, organ dysfunction, and malnutrition [150,151]. Significant correlations were discovered between the development of pressure sores and age and baseline skin integrity [152]. 

Pressure ulcers are classified into four stages: [153] 

Stage 1: The sore only affects the upper layer of skin, which turns red and warm.

Stage 2 The area is harmed as the sore digs deeper beneath the skin’s surface.

Stage 3: The sore has penetrated the fat tissue and formed a crater-like appearance.

Stage 4: As the sore penetrates deeper into the muscles and ligaments, a large wound forms.

In 2005 [154], the National Pressure Ulcer Advisory Panel (NPUAP) coined the term “deep tissue injury” to describe a pressure wound that conceals tissue injury beneath intact skin. The area looks like a deep bruise and is at risk of worsening into a high-stage pressure ulcer. 

Bed sore management may include the following:

Pressure relief: Removing pressure from the pressure sore site by repositioning the patient on a regular schedule, padding the pressure sore area, or using beds that use air to continuously shift pressure points [155]. 

Antibiotics: Use empirical broad-spectrum antibiotics first until culture and sensitivity results are obtained [156]. 

Dressings: There is no evidence that one dressing type is superior to another. The wound size, depth, shape, location, the presence of exudate and its volume, thr presence of tissue undermining, the type of tissue in the wound bed, and the surrounding skin condition all influence the dressing selection [157,158]. 

Gauze dressings containing hyaluronic acid and silver sulfadiazine will help keep the environment moist while also acting as a superficial debridement maneuver for the biofilm on the sore [159].Dressing with alginate: Alginate is a natural polysaccharide that is highly absorbent and is used to treat infected pressure ulcers with excessive exudate [160].Polyurethane foam has a porous structure that can be covered with multilayer silicone adhesive. It keeps wounds moist, provides thermal insulation, and effectively absorbs fluids into its air-filled spaces through capillary action. As a result, foam dressing is appropriate for many exuding wounds as well as bed ulcer prevention [160,161].Hydrocolloid dressings are made up of a hydrophilic and self-adhering colloid granule that has been coated with an external waterproof polyurethane. Colloid granules are typically made of methylcellulose, gelatin, or pectin, which combine to form a gel-like substance that adheres to the wound. They are indicated for wounds that are low to moderately exuding, granular, or necrotic. It is not appropriate for high exudate because it may cause accumulation around the wound site. Hydrocolloid dressings should also not be used on infected wounds due to their adhesive nature, which can harm the delicate surrounding skin [160,161].Hydrogel dressings are a promising type of dressing material due to their porous structure, biodegradability, ability to incorporate growth factors, and controlled release. They are ideal for dry wounds and over-granulation tissue because they are watery and gel-based [162].Silver-containing dressings: Because silver has bactericidal properties, they are ideal for infected wounds. However, they should be discontinued after infection clearance because silver is toxic to keratinocytes and fibroblasts [163].Honey-based dressings: Honey is a natural product rich in antioxidants, enzymes, and sugars, as well as having anti-inflammatory and antimicrobial properties. Honey has several benefits for wound healing, including accelerated dermal repair and epithelialization, angiogenesis promotion, immune response promotion, and infection reduction. Medical honey is available as a gel or paste, as well as in dressings [164].

Negative-pressure wound therapy: Applying negative pressure through a porous interface, such as foam, aids in the uniform distribution of negative pressure over the wound surface [165]. A transparent film covers the foam, allowing a vacuum to be created inside the wound [166]. The foam dressing can be easily customized to fit wounds of unusual shapes. Filtered non-contaminated air will flow inside, while potentially contaminated air will be sucked out of the wound. This aids in the reduction of inflammatory exudate and the promotion of granulation tissue [167,168]. It is appropriate for wounds with a high exudate as well as wounds adjacent to fecal or other body fluid flow, as its seal prevents wound contamination [155]. 

Debridement: Surgical removal of devitalized tissues, and pus drainage are essential for the treatment of pressure sores. This will create granulated tissue throughout the ulcer cavity that can heal later with epithelialization [169]. Reconstruction via primary closure or flap reconstructive surgery can be conducted later for wounds where the healing process is stuck or with full-thickness skin loss and exposure of muscle or bone [170]. 

### 6.4. Traumatic Lesions Due to Invasive ICU Procedures 

Invasive ICU procedures such as central venous catheterization, arterial catheterization, endotracheal intubation, chest tube insertion, or percutaneous tracheostomy can result in a variety of skin-related complications such as ecchymosis, infection, skin injury, or necrosis, as well as extravasation injuries [171,172]. 

### 6.5. Contact Dermatitis

Is brought on by allergens or irritants in tape, cleaning agents, or topical medications. It is characterized by erythema and vesico-bullous eruption, which may be exuding [173]. 

The priority is to identify and avoid the irritant that is causing the problem. Topical corticosteroids are an effective treatment that reduces inflammation quickly [174]. 

### 6.6. Skin Necrosis Associated with Disseminated Intravascular Coagulation

Skin necrosis caused by disseminated intravascular coagulation (DIC) typically manifests as tender, large irregular areas of purpura, particularly over the extremities, which can quickly enlarge into hemorrhagic bullae with subsequent necrosis [175]. 

## 7. Conclusions

The current review includes clinical aspects of the most common DMs that can occur in critically ill patients. Diagnosis in ICUs is more difficult if the patient is unable to express their pain, itching, or other symptoms due to sedation or neurological disorders.

Life-threatening DMs must be identified quickly and treated aggressively.

DMs associated with systemic diseases may aid in the diagnosis of underlying diseases.

During daily patient care, intensivists should anticipate and highlight DMs that may occur during critical illness and consult dermatologists as required.

A comprehensive algorithm for dealing with DMs in ICUs should be incorporated into routine quality care.

## Data Availability

Not applicable.

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
