# Peer review of "An Overview of Clinical Manifestations of Dermatological Disorders in Intensive Care Units: What Should Intensivists Be Aware of?"

_diagnostics, 2023, doi:10.3390/diagnostics13071290_

Round 1
Reviewer 1 Report
Comments to authors
Title
The title is concise and informative but still needs improvement to produce a good sound. The author is requested to revise the title.
Keywords
Please add some strong key words
Abstract
Complete, however I would suggest if author avoid the abbreviations in the abstract.
Introduction
Complete
Materials and methods
This portion provided a sufficient detailed methodology followed.
Results and Discussion
Results are clear concise and well presented. Some grammatical mistakes are present that need improvement.
References
Please ensure that every reference cited in the text is also present in the reference list. Follow the general guideline of the Journal for references.

Author Response
We would like to sincerely thank you, the editorial office and the reviewers for the time and effort taken to consider our manuscript. All the comments we received are highly appreciated and fully considered by us. Detailed response is in table below.
Sincerely,
Authors

Reviewer 2 Report
Very interesting article that can be very useful for intensivists.
However from a dermatological point of view it can be improved in some aspects to be more clear and precise
1) First consideration you cannot start with the abbreviation DMS it must be put in full first
2) In section 1.1 of pemphigus vulgaris some references to the sentences written by the authors are missing , when citing vaccines one cannot fail to consider the covid 19 vaccine which is a very current topic , the authors can cite this article which is inherent ( doi:10.1111/jdv.18302)
Also in this section the authors say " The cur- rent cornerstone treatment for pemphigus is local and systemic glucocorticoids" the reference is missing, they can use this ( doi:10.1159/000521712 )
3) In section 1.9
Treatment options usually include the use of drug combinations such as etanercept or infliximab (a biologic) together with cyclosporine or methotrexate (an immunosuppressant) prescribed by a dermatologist.
This phrase needs to be reshaped because it has no dermatologic basis; one does not prescribe biologics together with cyclosporine or mtx. One should at most follow a guideline in biologics , the authors can see treatment guidelines from this article
doi:10.2147/CCID.S364640
DOI:10.1111/jdv.14949
4) In section 3.3 when discussing the causes of vitiligo, the authors may also add this cause as reported by some works in the literature
doi:10.1111/dth.15102
DOI:10.1111/dth.15314
5) there are many typos, nonconforming characters and red underlines, the whole text needs to be revised and checked
6) Section 4.1 is very confusing; it needs to be revised entirely. It is better to write a single text without symbols . Also section 4.3
7) Another section should be added in the text for pityriasis rosea, which is another very frequent manifestation .
Etiology causes and treatments should be reported, in this case the covid 19 vaccine or the infection itself was the cause of several occurrences of this manifestation.
Here are some articles where authors can take their cues
- DOI: 10.1002/jmv.27535
- doi:10.1002/jmv.27672
Author Response

(The authors gave the same response as above.)

Round 2
Reviewer 2 Report
The authors have responded to all corrections.
The article is suitable for publication.